# Cost–Benefit of Real-Time Multiplex PCR Testing of SARS-CoV-2 in German Hospitals

**DOI:** 10.3390/ijerph20043447

**Published:** 2023-02-15

**Authors:** Roland Diel, Albert Nienhaus

**Affiliations:** 1Institute for Epidemiology, University Medical Hospital Schleswig-Holstein, Kiel, Airway Research Center North (ARCN), 24015 Kiel, Germany; 2Lung Clinic Grosshansdorf, Germany, Airway Disease Center North (ARCN), German Center for Lung Research (DZL), 22949 Großhansdorf, Germany; 3Institution for Statutory Accident Insurance and Prevention in the Health and Welfare Services (BGW), 22089 Hamburg, Germany; 4Institute for Health Service Research in Dermatology and Nursing, University Medical Center Hamburg-Eppendorf, 20246 Hamburg, Germany

**Keywords:** cost–benefit analysis, point-of-care, antigen testing, real-time reverse transcriptase polymerase chain reaction (RT-PCR), SARS-CoV-2, COVID-19

## Abstract

Background: The current Omicron COVID-19 pandemic has significant morbidity worldwide. Objective: Assess the cost–benefit relation of implementing PCR point-of-care (POCT) COVID-19 testing in the emergency rooms (ERs) of German hospitals and in the case of inpatient admission due to other acute illnesses. Methods: A deterministic decision-analytic model simulated the incremental costs of using the Savanna^®^ Multiplex RT-PCR test compared to using clinical judgement alone to confirm or exclude COVID-19 in adult patients in German ERs prior to hospitalization or just prior to discharge. Direct and indirect costs were evaluated from the hospital perspective. Nasal or nasopharyngeal swabs of patients suspected to have COVID-19 by clinical judgement, but without POCT, were sent to external labs for RT-PCR testing. Results: In probabilistic sensitivity analysis, assuming a COVID-19 prevalence ranging between 15.6–41.2% and a hospitalization rate between 4.3–64.3%, implementing the Savanna^®^ test saved, on average, €107 as compared to applying the clinical-judgement-only strategy. A revenue loss of €735 can be avoided when SARS-CoV-2 infection in patients coming unplanned to the hospital due to other acute illnesses are excluded immediately by POCT. Conclusions: Using highly sensitive and specific PCR-POCT in patients suspected of COVID-19 infection at German ERs may significantly reduce hospital expenditures.

## 1. Introduction

The severe acute respiratory syndrome COVID-19, caused by coronavirus 2 (SARS-CoV-2), first appeared in December 2019 in Wuhan, China with an accumulation of pneumonia and has since spread across the globe [1]. Clinical features of the disease include fever, headache, and cough, but more severe symptoms such as shortness of breath and respiratory failure have also been reported [2]. As of 30 October 2022, 627 million confirmed cases of SARS-CoV-2 infection and 6.5 million deaths have been reported from around the world [3]. More than 35.5 million cumulative cases have been reported to the German Robert Koch Institute (RKI) [4].

The Omicron variant was detected in South Africa in late November 2021 and is currently the dominant strain reported in Germany. The individual risk for hospitalization and death [5] for persons suffering with Omicron is less than that experienced with the previously prevalent Delta variant. The effective reproduction number for Omicron is, however, 3.8 times greater than that of Delta, and so Omicron leads to higher absolute hospitalization numbers in many countries, given the sheer numbers of persons infected [6].

Because Omicron is more transmissible than the Delta variant, a fresh surge is expected in autumn of 2022 to coincide with the beginning of the influenza and respiratory syncytial virus (RSV) season. The clinical picture of the Omicron variant is heterogeneous and—if a clinically severe course is present that leads to a visit to an emergency room—difficult to distinguish from influenza and RSV. Therefore, poor clinical specificity in the differentiation of respiratory infections can be assumed, and accurate and rapid identification of those infected with SARS-CoV-2 remains key to immediate clinical care and to the containment of the ongoing pandemic.

The current reference test used to establish SARS-CoV-2 infection worldwide is the real-time reverse transcriptase polymerase chain reaction (RT-PCR). These assays have nearly perfect sensitivity and specificity and are therefore well-suited as the “gold standard” for the diagnosis of clinically ill patients. However, the utilization of RT-PCT tests for immediate COVID-19 in hospitals raises substantial challenges: as they require ribonucleic acid (RNA) extraction, are dependent on the availability of PCR reagents, and have a relatively long turnaround time, RT-PCR tests are often performed in batches in clinical laboratories outside the hospital, necessitating specimen transport. Therefore, they usually have a time lag of one day before the test result becomes available. In Germany, currently 71.4% of all hospitals have eliminated their in-house laboratories [7].

In the meantime, several rapid real-time PCR tests are available whose use offers the potential for rapid identification of those individuals in the emergency setting who are not only infected, but infectious and are therefore at greatest risk of spreading the infection. One example is the Savanna^®^ Respiratory Viral Panel-4 (RVP-4, Quidel Corporation, San Diego, CA, USA), intended for use with the Savanna instrument for the simultaneous qualitative detection and differentiation of SARS-CoV-2, influenza A (Flu A), influenza B (Flu B), and respiratory syncytial virus (RSV). As results may be provided within 20 min without the need for upfront specimen preparation, this multiplex system can be used as POC test. Implementing such devices makes sense for three reasons: first, as COVID-19 patients have to be treated in isolation, fewer beds can be occupied overall. Previous payments for hospitals according to Section 21 (1b) of the German Hospital Act (KHG) and supply surcharges according to Section 21a of the KHG to compensate the loss of revenues when beds cannot be occupied as planned during the SARS-CoV-2 pandemic have expired as of 30 June 2022. PCR-POCT may help to prevent the hospital—if suspected COVID-19-infected patients test negative—from isolating such patients and blocking a second bed in the patients’ rooms at the hospital ward unnecessarily.

Second, rapid assessment of infectious COVID-19 is highly relevant to the management of scarce economic resources for another reason. Since 1 January 2004, hospital costs in Germany have been based on the German diagnosis-related groups (G-DRG) system, which assigns each incidence of COVID-19 pneumonia due to a COVID-19 case to the category E79C. This imposes a fixed “base rate” of payment for 13 days of treatment. If the hospital treatment exceeds the so-called “mean length of stay”, i.e., 6.9 days (as calculated mathematically by the DRG Institute for Hospital Reimbursement (InEK) using case-related data of its contracted hospitals [8]), then the G-DRG rate paid as reimbursement by the statutory health insurance (SHI) usually does not cover the costs incurred by the hospital. Accordingly, in treating COVID-19 patients covered by the SHI, hospitals should try to keep the duration of hospital stays as short as possible [9]. Whereas infected persons recovering at home may be de-isolated as early as 5 days after the onset of symptoms, according to the most recent guidelines of the German Robert Koch Institute (RKI) [10], isolation of hospital patients can be stopped and discharge initiated only if a lasting improvement in acute COVID-19 symptoms has been observed for >48 h and—in the case of pneumonia with a need for oxygen—a RT-PCR test is negative [11]. As the negative result of a RT-PCR test that is performed at the hospital is usually available one day earlier than that of the RT-PCR sent to an external laboratory, costs may be saved from the hospital´s perspective by an earlier discharge.

Third, pre-hospital laboratory PCR results are often not available for acute inpatient admissions. If no POCT is in place, a COVID-19 infection must be clinically ruled out in patients on the day of admission and the patient isolated on the ward until the (negative) result of the external RT-PCR test is made known the following day.

The aim of our calculations was to examine whether routine implementation of RT-PCR POCT in COVID-19 suspects visiting an ER leads to directly measurable economic advantages from the hospital perspective, taking as an example the Savanna^®^ Respiratory Viral Panel-4 under the assumption that all nasopharyngeal/nasal swabs of COVID-19 suspects are processed immediately. Using its performance characteristics [12], we compared the economic outcomes to those that occurred when conventional clinical judgement alone was initially used to confirm or exclude SARS-CoV-2 in patients deemed to have a combination of symptoms so serious as to warrant hospitalization. The hypothetical savings would come about thanks to earlier patient classification, in anticipation of a RT-PCR result, available only one day later.

## 2. Materials and Methods

### 2.1. Test System

The Savanna Instrument and test cassette are specifically designed for POCT in time-critical environments such as emergency departments. The test does not require any upfront specimen preparation, thus minimizing the risk of preanalytical mistakes. It provides a differentiated test result for Flu A, Flu B, RSV, and SARS CoV-2 in 20 min. The illuminated cartridge bay allows for monitoring of the process status from afar, eliminating unnecessary trips to the instrument. The test can process both direct swabs and specimens in transportation medium via its two specialized ports. A nasal or nasopharyngeal swab collected from the patient is transferred directly into the direct swab port of the test cassette, followed by breaking off and releasing the swab tip into the test cassette. After closing the lid of the dry swab port, the test cassette is transferred into the Savanna Instrument to start the analytical process. A second port is used for liquid medium, in which swabs reside during transport. A 250 µL aliquot of the transport liquid is pipetted into the test cassette via that port. The residual transportation medium can then be sent to and stored in the laboratory for research purposes.

### 2.2. Model Approach

Our model is parametrized by data on sensitivity and specificity of the Savanna^®^ Respiratory Viral Panel-4 [12] compared to the conventional clinical approach. For POCT, two scenarios were considered: in the first, all patients with one or more symptoms of a severe viral infection such as fever or chills, cough, shortness of breath, fatigue, muscle aches, strong headache, or sore throat visiting the ER of a hospital during the current COVID-19 pandemic were tested with the Savanna^®^ after using a nasopharyngeal/nasal swab. Depending on the severity of symptoms, a patient is hospitalized or discharged from the ER. In case of hospitalization, the patient is isolated from the moment of presumptive diagnosis after a positive Savanna test result until resolution of fever and respiratory symptoms, which currently takes a median length of 10 days [13]. Given the high sensitivity and specificity of the Savanna-RT-PCT (99.3% and 100%, see Appendix A for details), additional RT-PCR testing of the patient´s samples at external laboratories is not required in patients whose test is negative.

Due to the increased risk of thromboembolism associated with COVID-19 disease, a course of antithrombotic prevention using low molecular weight heparin at half the therapeutic dose and treatment with 6 mg dexamethasone is immediately started in all COVID-19 suspects who are admitted as inpatients from the ER because of the severity of their clinical condition [14].

In the alternative scenario, i.e., in the conventional clinical approach (versus Savanna^®^), the decision as to whether the present respiratory symptoms are caused by COVID-19 was made using symptom-based judgement, without rapid pre-testing. Thus, if hospitalization was required, the decision to isolate a COVID-19 suspect was only based on that clinical decision. In any case, a clinical sample in the form of a nasopharyngeal swab was taken from all COVID-19 suspects deemed to require hospitalization and sent out for external RT-PCR testing.

RT-PCR testing as done in external laboratories will generally have both sensitivity and specificity approaching 100%. Hence it serves to clarify whether or not the disease is due to SARS-CoV-2 and also flags and corrects false negative clinical classifications.

If the patient is not to be hospitalized, but discharged and sent home directly from the ER, SHI is charged for the costs of routine diagnostics (chest X-ray, routine laboratory values, physical examination, etc.) as well as, at least in part, the costs of the PCR-POCT, the latter following the the coronavirus test regulation (currently €32) [15].

As the costs of the Savanna^®^ testing in the ER fall to the hospital itself, these patients are included in our analysis.

For each patient ultimately hospitalized, an amount of €37.80 is reimbursed for RT-PCR according to the 3rd agreement of the German Hospital Finance Act (Krankenhausfinanzierungsgesetz, KHG) [16], irrespective of whether PCR testing has been performed in the hospital as POCT or in an external laboratory. We assume that the costs of external PCR testing billed by the laboratories and the amount of reimbursement cancel each other out. Accordingly, only the difference between the costs of initial Savanna testing, the swabs of which are taken in the ER, and the amount of reimbursement in Euros appears as a cost factor in our model.

Additional costs from the hospital perspective are the so called “opportunity costs”at the expense of the hospital that might occur as long as a potential COVID-19 patient is unneccesarily kept in isolation (see details below). This occurs in the cases of a false-positive clinical judgement or a false-positive PCR-POCT. Under the premise that most COVID-19 patients are accommodated in a twin-bedded room and that hospital wards in Germany during the COVID-19 pandemic are working at nearly full capacity, the economic losses caused by blocking the second bed are incurred by the hospital itself.

If a patient is isolated due to erroneous clinical judgement (no SARS-CoV-2 infection present), the isolation can be ended as soon as the report of the negative laboratory RT-PCR result is available the next day. It is assumed that the administration of low-molecular-weight heparin and dexamethasone is continued until discharge if SARS-CoV-2 infection is confirmed by external PCR. In the case of a negative PCR result, that medication is dropped immediately. Thus, patients falsely suspected of having COVID-19, by whatever means, end up being isolated and receiving antithrombotic prevention and dexamethasone for one day.

In the case of a false-positive Savanna^®^, the errant result goes unnoticed (as a false clinical judgment would not), because no external PCR test is performed. Under the worst case assumptions, patients that falsely tested positive would end up being isolated for 10 days [13], the current median duration of hospital stay COVID-19 patients in Germany, and ineffectively receive low-molecular-weight heparin and dexamethasone. Early release from isolation is, in these cases, out of the question.

In Germany, according to the current recommendations of the Robert Koch Institute, discontinuing isolation prior to discharge relies on a negative test result, at least in case of a severe course of the disease preferably RT-PCR [11]. It can be expected that, by performing a PCR-POCT, patients can be discharged one day earlier than forseen by the DRG, sparing the assumed delay that external RT-PCR testing imposes. As the hospital receives a fixed DRG flat rate in any case, this would result in an economic benefit to the hospital.

Our model does also takes into account the effects of COVID-19 transmission to unvaccinated health care workers by COVID-19 sufferers who have gone undetected and not been isolated due to false clinical judgement or a false-negative POCT result. Since 15 March 2000, all health care workers in German hospitals are required to be vaccinated [17], but current vaccines provide only limited long-term protection, especially against infections with Omicron [18]. For this, we have incorporated a secondary attack rate leading to sick days for hospital workers, the costs of which, under the German system, are borne by the hospital. For purposes of simplification, in our model, only one fully vaccinated health care worker is assigned to an unisolated patient.

### 2.3. Model Structure

The decision tree simulates the outcomes of two management strategies in the ER of a German hospital in a hypothetical cohort of 1000 adult patients attending the ER with acute moderate-to-severe respiratory infection and suspicion of COVID-19 (Figure 1). Costs from the hospital perspective were compared as described above: (1) Savanna^®^ POCT without additional RT-PCR testing or (2) empiric clinical investigation without POCT in the ER, but with subsequent external PCR testing due to the low clinical sensitivity and specificity of clinical judgement on the presence of SARS-CoV-2 infections. In mild cases who will be sent home again from ER according to clinical judgement, an external RT-PCR is not required; in contrast, in the Savanna–POCT strategy, all patients are tested, whether hospitalized or not.

Total costs of outcomes were simulated for each study arm, including (1) medical cost of POCT with the Savanna^®^ Multiplex RT-PCR (Scenario1), (2) medical costs of external RT-PCR testing performed after hospitalization (Scenario 2), (3) opportunity costs due blocking a twin-bed reimbursement for one day of hospital stay, (4) reimbursement per day of hospital stay within the fixed payment DRG period, (5) sick pay costs at the expense of the hospital if staff members are secondarily infected by hospitalized but unrecognized COVID-19 patients, and (6) medical costs of initial treatment with enoxaparin and dexamethasone in hospitalized patients (Figure 1).

We used TreeAge Software (TreeAge Inc., Williamstown, MA, USA) for model building and analysis and examined our inputs over a wide range in sensitivity analyses to identify influential factors that would alter the base-case findings. First, univariate sensitivity analysis was performed using all variables to examine the extent to which our calculations were affected by varying selected assumptions. Variation was done using either (a) the lower and upper bounds of a parameter´s standard deviation or (b) those of its 95% confidence interval. Where these were not applicable, our model simply caused parameter values to vary by ±20% of the base-case value according to international practice, unless stated otherwise.

Furthermore, and in order to capture the interactions between multiple inputs, we conducted a probabilistic sensitivity analysis (PSA) by assigning an appropriate statistical (probability) distribution for all parameters, randomly drawn in a 2nd order Monte Carlo simulation (n = 1000). We chose uniform distributions for all cost parameters and PERT distributions for all frequencies and probabilities. All costs are reported in 2022 Euros (€).

### 2.4. Model Input

The figures for the other epidemiological, laboratory, and economic parameters are listed in Table 1; their origins are described in detail in the Appendix A.
ijerph-20-03447-t001_Table 1Table 1Input for cost–benefit analysis.Variables CategoryVariable NameDistribution *Value (Base Case)Relative Change (Range)Reference (in Appendix A)Prevalence of COVID-19COVID_prevPERT0.1560.079–0.412[19]Additional revenue per day due to earlier dischargecRev_day_POCTuniform€329.42±20% (€263.54–€395.30)Calculated using data from Institute for the Hospital Remuneration System (InEK) [16]Specificity of Savanna^®^ testingSavanna_COVID_ specuniform195% CI (0.9542–1) [20]Opportunity costs due to blocking twin bedcOpp_POCTuniform€734.53±20% (€587.62–€881.44)Calculated from InEK data [16]Clinical probability of correctly excluding SARS-CoV-2 Clin_spec_ COVIDPERT0.68395% CI (0.60–0.758)[21]Clinical sensitivity of diagnosing SARS-CoV-2 infection Clin_sens_ COVIDPERT0.80695% CI (0.729–869)[21]Costs of enoxaparin per daycAntithromb_dayuniform€6.91±20% (€5.53–€8.30)Rote Liste [Red List] 2022Costs of dexamethasone per daycDexa_dayuniform€1.3572±20% (€1.0857–€1.6286)Rote Liste [Red List] 2022Costs of Savanna^®^ PCRcSavanna_COVIDuniform€40±20% (€32–€48)As declared by manufacturerSensitivity of Savanna^®^ testingSavanna_COVID_ sensPERT0.993195% CI (0.9617–0.9988)[20]Secondary cases in HCW due to one unknown COVID-19 casesec_COVID_ HCWPERT0.024895% CI (0.0085–0.0704)[22]Costs of productivity loss per daycPL_dayuniform€170.90+20% (€205.8)Calculated from [23]Number of days of health care workers out of work due to COVID-19sick_daysuniform15+12 (27)[24,25]Probability that hospitalization is requiredpHospPERT0.604495% CI (0.5645–0.6429) [PSA: 0.043–0.6429][26]Length of hospital stay (median)dHospPERT105–19 (IQR)[27]Costs of RT-PCR performed in external laboratorycRT-PCR_extuniform€37.8+50% (€56.70)Nationwide laboratory inquiry***** in probabilistic sensitivity analysis (PSA).
ijerph-20-03447-t002_Table 2Table 2Results of the base-case analysis (no confirmation of Savanna^®^ results by external RT-PCR).Base-Case AnalysisComparatorsMean CostPer Patient (€)Incremental Cost (€) *Absolute Cost Savings (€) COVID-19 patients prior to hospitalizationSavanna^®^ Multiplex RT-PCR −25.310−25.31Conventional approach123.06148.37
* Incremental cost denotes the increase in total costs resulting from using the conventional approach alone versus Savanna^®^ Multiplex RT-PCR POCT.

## 3. Results

In the base-case analysis, utilizing the Savanna^®^ Multiplex RT-PCR in COVID-19 patients is, on average, €148.37 less costly per patient tested in the ER as compared to the conventional clinical approach (see Table 2), although no POCT result was re-checked by external PCR. Included in this amount is a cost savings of €25.31 in absolute terms per tested patient in favor of the hospital. For patients who end up being discharged from the ER following examination, the costs for initial Savanna^®^ testing, which are only partly reimbursed to the hospital by the SHI, are considered here, whereas the incurred external laboratory costs following the conventional clinical judgement—in contrast to the POCT—are de facto fully covered.

In univariate analysis, the specificity of the Savanna^®^, i.e., its ability to accurately exclude a COVID-19 infection (see Table 3 and the graphical representation in Figure 2), represents 81% of the total uncertainty. Reducing the base-case value of 100% assumed specificity to 95.4% (worst case) results in higher costs compared to the strategy of clinical judgement of €22.74, at the expense of the hospital. Here, the threshold value at which the costs of the two strategies are equal is a specificity of Savanna^®^, at 96.14%.

Far behind this follows the impact of changing the assumed specificity of clinical judgement: reducing the base case value of 68.3% to 60.0% (worst case) results in a further cost savings of €31.91 (€148.37 minus €180.28) on top of the €153.71, while an increase to 75.8% diminishes the saving by €28.84 to €119.53. This is revealed by our univariate sensitivity analysis, in which all variables included in the decision analysis change between plausible extreme ranges (Table 3). The opportunity cost of blocking a twin bed to the disadvantage of the hospital is the third important component: decreasing by 20% the assumed opportunity cost reduces the amount of cost saving by €25.54 to €122.83.

A decreasing probability that hospitalization is required has less influence on the economic outcome: when decreasing the assumed hospitalization rate to the lower bound estimate of the 95% confidence interval, i.e., by 9.3%, from the base case value of 60.44%, the cost savings decrease only to €137.91. Even if the hospitalization rate due to an unexpectedly high number of patients with milder symptoms were only 4.3%—and 95.7% of the ER patients would be sent home again after being tested with the Savanna^®^—the relative cost savings per patient would still be €1.27 in favor of the POCT strategy (data not shown).

A decrease in the number of COVID-19 cases in the ER, i.e., a lower level of prevalence, lead to almost equal changes in costs for both strategies: even under worst-case assumptions, where only 7.9% of all patients with respiratory symptoms reporting to an ER turn out to be COVID-19 cases, a cost saving of €143.81 in favor of the hospital remains.

An increase of the costs of performing the Savanna^®^ POCT in the ER by 20%, or €8, diminishes the cost saving by nearly the identical amount (€8.75). Although the multiplex PCR test is clearly more expensive than SARS-CoV-2 antigen tests, which have a lower sensitivity and specificity, doubling the current amount from €40 to €80 does not lead to a reversion of the relative cost savings, but the savings are reduced to €104.62 (data not shown). Vice versa, by decreasing the price of the test by 20%, i.e., so that the hospital would have to pay €5.8 less than the €37.8 reimbursed per tested patient, the cost savings in favor of the hospital would amount to €157.12 (see Table 3).

Variations of all other parameters do not or do only marginally change the absolute amount of expenditures in favor of the hospital.

In probabilistic sensitivity analysis (PSA), where the results are based on random-sampling and therefore differ from those of the univariate analysis, performing Savanna^®^-POCT on each patient prior to hospitalization reduces the costs that occur when COVID-19 suspects are isolated based only on the conventional clinical approach, by €106.79 (see Table 4). Of note, testing with Savanna^®^ Multiplex RT-PCR is in 99.90% of cases less expensive than the initial clinical approach, even when considering a frequency of hospitalization of only 4.3% as the lower bound in the PSA. Furthermore, as the net costs for utilizing the Savanna^®^ are less than zero (- €0.38), the use of the test offsets the costs associated with it.

Most of this savings is due to the fact that the overwhelming majority of hospitalized patients (98.9%) who were tested correctly with the highly sensitive Savanna^®^ and are now tested the second time at the end of their stay can be discharged immediately. Thus, €325.02 can be saved in each of those patients in favor of the hospital, while with the alternative strategy where no POCT is available, the negative PCR test results will be reported no earlier than the following day. Furthermore, the proportion of initial unnecessary bed-blocking is nearly forty-fold higher (31.8% vs. 0.8%) with conventional clinical judgement than with the Savanna^®^ Multiplex RT-PCR. As this mistake can be corrected only 1 day later, when the result of the external RT-PCR is available, the cost difference between the two strategies, with respect to opportunity costs—weighted by the proportion of 81.4% of patients who were not infected with SARS-CoV-2—is €183.23 in favor of the Savanna^®^ Multiplex RT-PCR test.

## 4. Discussion

Newer real-time PCR-POC tests such as the Savanna^®^ Multiplex RT-PCR, which can claim perfect specificity, may very rapidly and reliably exclude viral presence in a patient of transmissible COVID-19. Therefore, they offer the potential to avoid unnecessary isolation that occurs to a great extent under the conventional clinical approach. Thus, in the PSA of our model, the routine implementation of a PCR-POCT for possible COVID-19 patients before being moved from the ER for admission to a German hospital ward is almost consistently (99.9% of cases) about €107 less expensive than the conventional symptom-based judgement, for which the RT-PCR testing results are available only from the following day. Furthermore, PCR-POCT allows a planned discharge from the hospital, for which a final negative testing is required according to the current guidelines, one day earlier compared to waiting for the results of an external laboratory. Thus, the bed occupied by a COVID-19 patient is freed up a day earlier, and the hospital saves about €329 as the DRG flat reimbursement rate per COVID-19 stay is paid without regard to the length of the stay. Of note, this ranking is not dependent on changes in the prevalence of COVID-19 in such patients, as long as the COVID-19 prevalence in ER patients does not exceed 41.2%. The ranking is also not dependent on the rate of hospitalizations unless and until it falls below 4.3%. There have been complex attempts to better predict the presence of COVID-19 by create artificial intelligence (AI) programs that process clinical data as well as imaging techniques. Xia et al. [28] describe that when considering 52 clinical and laboratory coefficients, e.g., disseminated intravascular coagulation, d-dimer, procalcitonin, enlarged lymph nodes, or rhabdomyolysis together with CXR features, sensitivity increased to 94% and specificity to 75%. However, the whole bundle of information required is hardly available in the setting of an ER before deciding whether a possible COVID-19 patient should be hospitalized or not.

Additional cost savings in favor of the hospital arise when patients are admitted to the hospital as acute patients for causes other than SARS-CoV-2 infections. Meanwhile, as a planned pre-stationary PCR test is de facto not possible here, such patients must first be isolated in a bed on the ward to await the—hopefully negative—test result from an external laboratory on the following day. The unnecessary blocking of the second bed in a twin-bedded room leads to an opportunity cost of € 735 for that day. In contrast, with the help of a POC test system, the quarantining of SARS-CoV-2 infection can take place in the waiting area prior to admission to the designated ward. If the acute hospital has high patient throughput, considerable revenue losses can be avoided in such situations, which often force snap clinical decisions. To date, pivotal studies quantifying the magnitude of these losses and the degree to which POCT economically challenges the conventional approach are not available.

Furthermore, POCT, by the example of the Savanna^®^, may significantly facilitate the organization of work in the ER. Preanalytics, e.g., transferring swab material into viral transport media, is not required or reduced to a minimum. The end of the 20-min testing process is indicated by the flash of an illuminated LED ring at the device that can easily be seen from a distance of up to 50 m. Thus, personnel are freed to pursue other tasks, undistracted. In addition, the availability of prepackaged testing may save physician time for handling cases in the ER, as the need to query the patient about symptoms and to examine them once test results are in hand becomes less critical. Although it is difficult to produce a monetary estimate of the time value thus saved, it is safe to assume that reliable, sensitive testing as suggested would help to reduce crowding in the ER.

Our study has some limitations that must be kept in mind when interpreting its results. As always, the general limitation of a single-center economic model that cannot depict the reality of bed capacity utilization for each hospital deserves consideration, as does the local SARS-CoV-2 infection prevalence among exposed health care workers. Therefore, to validate our estimates, prospective cost studies, preferably with a multicenter study design, are required. Furthermore, our calculations refer only to hospitals that must send samples to an external laboratory for COVID-19 testing and wait for the report. Hospitals that have a laboratory department at their disposal that already conducts high quality RT-PCR tests while the patients are waiting in the ER, even during weekend and at night, will probably not benefit by COVID-19 POCT. Another limitation of our model is that the need to be hospitalized always depends on the clinical discretion of the physicians in the ER and cannot be predicted in individual cases. However, as a high spread may be assumed, the basic percentage of 60.4% was varied in a broad sensitivity analysis.

Additionally, our model probably underestimates the economic benefits of the Multiplex panel. Not only is an increase of Omicron infections expected in the fall and winter season, but flu and RSV as well. Infections borne by this triple threat could simultaneously be detected (or excluded) by the panel to guide isolation management. Assessing such aggregated cost benefits, however, was not the target of our calculations.

## 5. Conclusions

The utilization of the Savanna^®^ Multiplex RT-PCR test, being representative of new RT-PCR POC tests, is likely to reduce hospital-related costs in cases of suspected COVID-19 in German emergency departments. As such, PCR-POCT can reduce costs from the hospital´s perspective and allow resources to be allocated for other matters. Prospective clinical studies should be undertaken to further evaluate its economic advantages and include the simultaneous detection of flu and RSV cases in the immediate future.

## Figures and Tables

**Figure 1 ijerph-20-03447-f001:**
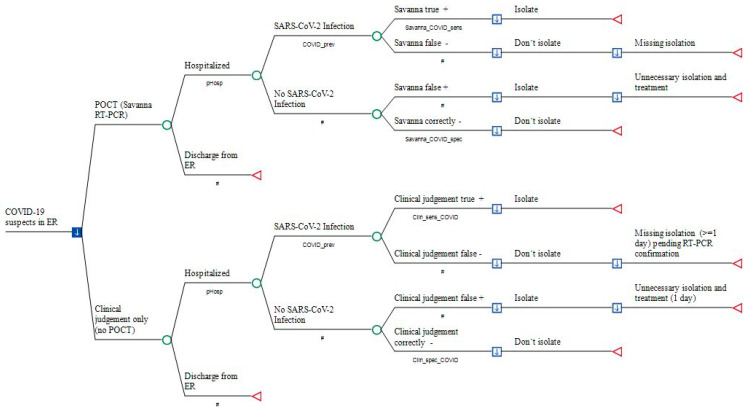
Point-of-Care antigen Testing (POCT) versus the conventional approach in COVID-19 suspects prior to hospitalization. Legend: a decision node (square) indicates a choice facing the decision-maker or the consequences of a decision. Branches from a chance node (circles) represent the possible outcomes of an event; terminal nodes (triangles) denote the endpoints of a scenario and are assigned in Table 2. Savanna_COVID_sens: sensitivity of Savanna^®^ Multiplex RT-PCR; Savanna_COVID_spec: specificity of Savanna^®^ Multiplex RT-PCR testing; Clin_sens_COVID: sensitivity of diagnosing SARS-CoV-2 infection; Clin_spec_COVID: probability of correctly excluding SARS-CoV-2. #: Complementary probability (all probabilities of chance node’s branches to sum to 1.0); +: positive; −: negative.

**Figure 2 ijerph-20-03447-f002:**
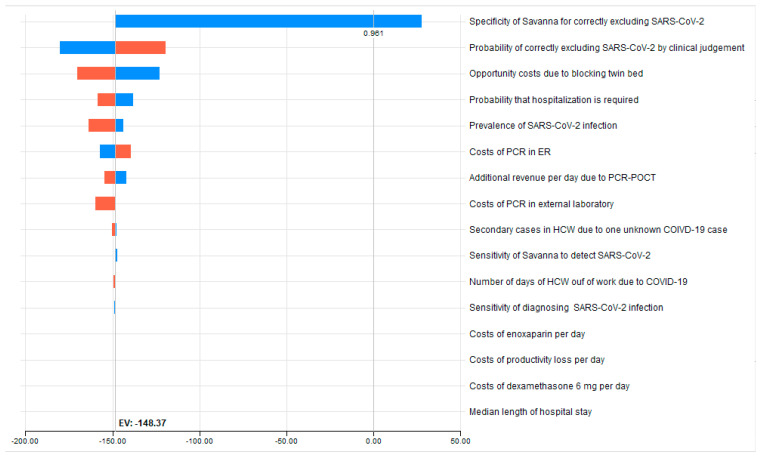
Tornado sensitivity analysis dashboard. The red and blue bars in the Tornado graph represent individual one-way sensitivity analysis performed for each variable. They include a vertical dotted line indicating the expected value (saving of €148.37) by utilizing the Savanna^®^.

**Table 3 ijerph-20-03447-t003:** Tornado diagram * (Savanna^®^ PCR-POCT versus the conventional clinical approach).

Variable Name	Variable Description	Lowest Value	Basecase Value	Highest Value	Savings at Lowest Value (€)	Savings at Highest Value (€)	Spread (€) ^Ƭ^	Risk % ^¥^	Cum Risk %	Threshold Variable Value
Savanna_COVID_ spec	Specificity of Savanna for correctly excluding SARS-CoV	0.9542	1	1	−148.37	27.74	176.11	0.808	0.808	0.9614
Clin_spec_ COIVD	Probability of correctly excluding SARS-CoV-2 by clinical judgement	0.6	0.683	0.758	−180.28	−119.53	60.75	0.096	0.904	-
cOpp_POCT (€)	Opportunity costs due to blocking twin bed	587.62	745.53	881.44	−122.83	−170.35	47.51	0.059	0.963	-
pHosp	Probability that hospitalization is required	0.5645	0.6044	0.6429	−137.91	−158.46	20.54	0.011	0.974	-
COVID_prev	Prevalence of COVID-19	0.079	0.156	0.412	−163.52	−143.81	19.70	0.010	0.984	-
cSavanna_COVID (€)	Costs of Savanna POCT in ER	32	40	48	−157.12	−139.62	17.50	0.008	0.992	-
cRev_day_ POCT (€)	Additional revenue per day due to POCT	263.54	329.42	395.30	−142.20	−154.54	12.34	0.004	0.996	-
cRT_PCR_ext (€)	Costs of PCR in external laboratory	37.8	37.8	56.7	−159.79	−148.37	11.42	0.003	1.000	-
sec_COVID_ HCW	Secondary cases in HCW due to one unknown COVID-19 case	0.0085	0.0248	0.0704	−150.43	−147.63	2.80	0.000	1.000	-
Savanna_COVID_ sens	Sensitivity of Savanna to detect SARS-CoV-2	0.9617	0.9931	0.9988	−147.21	−148.58	1.37	0.000	1.000	-
sick_days	Number of days of HCW out of work due to COVID-19	15	15	27	−149.27	−148.37	0.90	0.000	1.000	-
Clin_sens_ COVID	Sensitivity of clinically diagnosing COVID-19 if present	0.729	0.806	0.869	−148.83	−147.99	0.84	0.000	1.000	-
cAntithromb_day	Costs of enoxaparin per day	5.53	6.91	8.3	−148.59	−148.15	0.45	0.000	1.000	-
cPL_day (€)	Costs of productivity loss per day	170.9	170.9	205.8	−148.60	−148.37	0.23	0.000	1.000	-
cDexa_day	Costs of dexamethasone per day	1.09	1.36	1.6284	−148.41	−148.32	0.09	0.000	1.000	-
dHosp	Median length of hospital stay	5	10	19	−148.37	−148.37	0.00	0.000	1.000	-

* One-way sensitivity analyses of all model variables arranged in order, with the variable with the biggest impact at the top and the variable with the smallest impact at the bottom. ^¥^ Risk%: this is a measure of how much of the total uncertainty is represented by the respective variable. The Risk% values sum to 1.0 across all the variables. ^Ƭ^ Highest cost saving minus lowest cost saving in €.

**Table 4 ijerph-20-03447-t004:** Results of the probabilistic sensitivity analysis (Monte Carlo simulation).

Probabilistic Sensitivity Analysis	Comparators	Mean Cost Per Patient (€)	Standard Deviation (±SD)	Incremental Cost (€) *
COVID-19 patients prior to hospitalization	Savanna^®^ Multiplex RT-PCR	−0.38	25.22	−0.38
	Conventional approach	106.40	24.95	106.79

* Incremental cost denotes the increase in total costs resulting from using the conventional approach alone versus Savanna^®^ Multiplex RT-PCR POCT.

## Data Availability

See Appendix A.

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
