# Peer review of "Cost–Benefit of Real-Time Multiplex PCR Testing of SARS-CoV-2 in German Hospitals"

_ijerph, 2023, doi:10.3390/ijerph20043447_

Round 1

Reviewer 1 Report

This is a well done but quite conventional analysis of the cost of using a rapid COVID test on ER and admitted patients versus less accurate clinical judgment, and the cost savings from lower quarantine expenses if hospitalized from the more accurate test, from the viewpoint of hospital costs.  I am unclear whether this is a good fit for IJERPH but, if it is, the paper is of value.  I suggest some modest revisions for clarity and exposition purposes.

1.      It is not clear at the outset (page 3) which ER patients are going to be tested.  Later (page 5) this is clarified as only those with respiratory symptoms suggestive of COVID.  It would be useful to explore or at least comment on a strategy of PCR testing of all ER patients.  In either case, there would be benefits to patients or perhaps the insurer even if the patient is sent home from the ER if COVID status were established.

2.      Would patients testing positive in the ER be more likely to sent home for isolation rather than admitted?  Saving the cost of a hospital admission (even if it would have been (barely) justified without COVID) could amount to a large amount. In New York in early days of COVID patients were sent to nursing homes if infected rather than admitted—but this turned out to be a mistake because they spread the virus.  How the model treats this crucial change in potential behavior could be clarified.

3.      I was unclear how the crucial “cost” (actually price) of the test was treated.  German hospitals are reimbursed for PCR tests but there can be a shortfall between payment to the hospital and the price the hospital pays.  Since this difference will be much smaller than the price of the test, which is the price or cost to be included in the Cost Benefit analysis.  If the hospital was paid more than the cost of the test, also possible, how would its increase in net revenue be treated?

4.      The sensitivity analysis is done only with reference to results from the simulation not with reference to what might be happening in reality.  The most obvious issue is the effect of a lower price for the PCR test.  The 40 euro price attributed to the Savanna test studied seems high to me relative to what might also be available in the market.  If competition among test sellers and careful hospital procurement reduced the price of the test, that would presumably increase the net savings but by how much?  The other parameter that might be considered is the prevalence of COVID in the community.  If it is low enough, testing will not be cost beneficial.  How low would it have to go?

5.      Availability of prepackaged testing presumably will save on physician time in treating the patient in the ER since there would be less need to query the patient about symptoms and examine for them.  It is hard to get a money estimate of the value of time saved but it might help to reduce crowding in the ER.  Is this considered?

6.      Language should be edited.  “The” for “die”; “errant” for “irrant”, not “Far behind falls…”

Author Response

We are appreciative of the reviewers’ very useful input. We have revised the manuscript according to their suggestions, to the degree possible, and prepared a point-by-point response. Altered or added passages are marked red in the revised version. We hope that our manuscript now meets your expectations for publication in IJERPH.

Reviewer 1:

1. It is not clear at the outset (page 3) which ER patients are going to be tested.  Later (page 5) this is clarified as only those with respiratory symptoms suggestive of COVID.  It would be useful to explore or at least comment on a strategy of PCR testing of all ER patients.  In either case, there would be benefits to patients or perhaps the insurer even if the patient is sent home from the ER if COVID status were established.

Reply: Thank you for this comment, you are right!  To clarify what “COVID-19 suspects” are, we replaced the term ”COVID-19 patients” in the third line of the “Model approach” section  by “patients with one or more symptoms of a severe viral infection, such as fever or chills, cough, shortness of breath, fatigue, muscle aches, strong headache or sore throat, coming to the ER….”

2. Would patients testing positive in the ER be more likely to sent home for isolation rather than admitted? Saving the cost of a hospital admission (even if it would have been (barely) justified without COVID) could amount to a large amount. In New York in early days of COVID patients were sent to nursing homes if infected rather than admitted—but this turned out to be a mistake because they spread the virus. How the model treats this crucial change in potential behavior could be clarified.

Reply: Thank you for emphazising this question. As stated in the first para of the “Model approach”, patients testing positive for COVID-19 (or simply suspected to have COVID-19) will be hospitalized according to the severity of their symptoms; patients with mild symptoms will be sent home.

 Whether systems are so severe as to warrant hospitalization is a matter of clinical discretion on the part of the ER physicians cannot be predicted in individual cases. We have therefore based the model on the observed frequency of hospital admissions for suspected COVID-19 infection in Germany, which - as explained in more detail in the Supplement – has been reported to be 60.44%. As one must assume a high spread in practice, this percentage was varied in a broad-range sensitivity analysis, using lower and upper bounds of 4.3% and 64.3%, respectively.  To explain the problem and clarify our approach, we have now added a sentence in the Discussion section regarding  "limitations".

3. I was unclear how the crucial “cost” (actually price) of the test was treated. German hospitals are reimbursed for PCR tests but there can be a shortfall between payment to the hospital and the price the hospital pays.  Since this difference will be much smaller than the price of the test, which is the price or cost to be included in the Cost Benefit analysis.  If the hospital was paid more than the cost of the test, also possible, how would its increase in net revenue be treated?

Reply: Thank you, we have now added a sentence in the Result section to clarify the consequences for the hospital when the price of the test decreases by 20% in favor of the hospital. In practice, tests priced beyond the limits suggested by the reimbursement schedule are not commercially viable in Germany, so the situation in question need not be considered, in our opinion.

4. The sensitivity analysis is done only with reference to results from the simulation not with reference to what might be happening in reality. The most obvious issue is the effect of a lower price for the PCR test. The 40 euro price attributed to the Savanna test studied seems high to me relative to what might also be available in the market.  If competition among test sellers and careful hospital procurement reduced the price of the test, that would presumably increase the net savings but by how much?  The other parameter that might be considered is the prevalence of COVID in the community.  If it is low enough, testing will not be cost beneficial.  How low would it have to go?

Reply: Thank you for this comment. For our model-based cost-benefit analysis we used the most recent statistical input in order that the model approximate as closely as possible today’s reality. Indeed, the price of 40 Euro is much higher than that of an antigen test. Antigen POC tests are today in use in some German hospitals ERs, but they lack the higher sensitivity and a slightly higher specificity of the Savanna test. We have now added a corresponding half sentence in the Result section.

Also, we anticipate the market introduction of comparable tests by competing companies, and have therefore considered the effect of a market-driven drop in price of 20% (see our reply regarding 3.).

With respect to the prevalence of COVID 19, it can be seen that in our base case analysis, the POC-PCR test remains a cost saving even when prevalence was as low as 7.9%. We had already stated in the result section: “A decrease in the number of COVID-19 cases in the ER, i.e. a lower level of prevalence, lead to almost equal changes in costs for both strategies: Even under worst-case assumptions, where only 7.9% of all patients with respiratory symptoms reporting to an ER turn out to be COVID-19 cases, a cost saving of €143.81 in favor of the hospital remains.” We did not want to include speculation in our model as to whether or -if so- how quickly, COVID-19  prevalence may continue to drop. In all probability, Covid testing will not be performed in ER at all if a COVID-19 case becomes a rare event in the future.

5. Availability of prepackaged testing presumably will save on physician time in treating the patient in the ER since there would be less need to query the patient about symptoms and examine for them. It is hard to get a money estimate of the value of time saved but it might help to reduce crowding in the ER. Is this considered?

Reply: Thank you very much for this important comment. Indeed, although a thorough history of a patient's symptoms is always necessary in the emergency department, testing could enhance the processing of patients. We have now taken up this point directly and added a corresponding sentence to the discussion:

“ In addition, what should be kept in mind is that availability of prepackaged testing may well save on physician time in handling cases in the ER, as the need to query the patient about symptoms and to examine for them, once test results are in hand, becomes less critical.  Although it is difficult to get a monatary estimate of the value of time thus saved, it is safe to assume but reliable, sensitive testing as suggested would help to reduce crowding in the ER.”

6. Language should be edited. “The” for “die”; “errant” for “irrant”, not “Far behind falls…”

Reply: Thank you, the typing errors were corrected and the manuscript seen again by a native speaker.

Reviewer 2 Report

Model approach, page 3, last para:

“In the alternative scenario (versus Savanna®), i.e. in the conventional clinical approach, the decision as to whether the present respiratory symptoms are caused by COVID-19…”

This sentence is somewhat imprecise; here the alternative should come first. The sentence should read: “In the alternative scenario, i.e. in the conventional clinical approach (versus Savanna®), the decision as to whether the present respiratory symptoms are caused by COVID-19…”

Page 4, 3rd para: “For each patient are ultimately hospitalized”. The "are" is redundant and should be deleted.

Page 4, 6th para: “Additional costs from the hospital perspective are the so called “opportunity costs”. Costs for whom? It should be added: opportunity costs at the expense of the hospital…

Page 4, 8th para:” Under the worst case assumptions, patients that falsely tested positive would end up being isolated for 10 days…”  

That the number of isolation days (10 days) in the hospital is the current German average is unfortunately only presented to the reader in the next paragraph. I suggest to insert this already here and to change the sentence in the next para accordingly. For example, the sentence could read: is “Under worst case assumptions, patients that falsely tested positive would end up being isolated for 10 days [13], the current median duration of hospital stay COVID-19 patients in Germany, and ineffectively receive… 

The sentence in the next para could be shortened: “It can be expected that, by performing a PCR-…”POCT, patients can be discharged one day earlier than forseen by the DRG, sparing the assumed delay…”

Results, line 33: “Fluctuations of the number of COVID-19 cases in the ER, i.e. a lower level of preva- lence, lead to…” “Fluctuations” is not the correct word here, because the authors describe only the consequences of a decreasing number of COIVD-19 cases. I suggest replacing “Fluctuations of ” by “A decrease in the number of COVID-19 cases…”

Discussion, line 79: “€ 107 less expensive“. Delete the superfluous space between € and 107.

Discussion, line 85: “about €329 as DRG flat reimbursement rate per COVID-19 stay is paid without regard to 85 the length of the stay.”  Insert a “the” on front of “DRG flat rate”

Author Response

We are appreciative of the reviewers’ very useful input. We have revised the manuscript according to their suggestions, to the degree possible, and prepared a point-by-point response. Altered or added passages are marked red in the revised version. We hope that our manuscript now meets your expectations for publication in IJERPH.

Reviewer 2:

Model approach, page 3, last para:

“In the alternative scenario (versus Savanna®), i.e. in the conventional clinical approach, the decision as to whether the present respiratory symptoms are caused by COVID-19…”

This sentence is somewhat imprecise; here the alternative should come first. The sentence should read: “In the alternative scenario, i.e. in the conventional clinical approach (versus Savanna®), the decision as to whether the present respiratory symptoms are caused by COVID-19…”

Page 4, 3rd para: “For each patient are ultimately hospitalized”. The "are" is redundant and should be deleted.

Page 4, 6th para: “Additional costs from the hospital perspective are the so called “opportunity costs”. Costs for whom? It should be added: opportunity costs at the expense of the hospital…

Page 4, 8th para:” Under the worst case assumptions, patients that falsely tested positive would end up being isolated for 10 days…” 

That the number of isolation days (10 days) in the hospital is the current German average is unfortunately only presented to the reader in the next paragraph. I suggest to insert this already here and to change the sentence in the next para accordingly. For example, the sentence could read: is “Under worst case assumptions, patients that falsely tested positive would end up being isolated for 10 days [13], the current median duration of hospital stay COVID-19 patients in Germany, and ineffectively receive…

The sentence in the next para could be shortened: “It can be expected that, by performing a PCR-…”POCT, patients can be discharged one day earlier than foreseen by the DRG, sparing the assumed delay…”

Results, line 33: “Fluctuations of the number of COVID-19 cases in the ER, i.e. a lower level of preva- lence, lead to…” “Fluctuations” is not the correct word here, because the authors describe only the consequences of a decreasing number of COIVD-19 cases. I suggest replacing “Fluctuations of ” by “A decrease in the number of COVID-19 cases…”

Discussion, line 79: “€ 107 less expensive“. Delete the superfluous space between € and 107.

Discussion, line 85: “about €329 as DRG flat reimbursement rate per COVID-19 stay is paid without regard to 85 the length of the stay.”  Insert a “the” on front of “DRG flat rate”

Reply: Thank you, we have revised all sentences as suggested.

Reviewer 3 Report

Report.

The article assesses the cost-benefit of implementing PCR POC COVID-19 testing (Savanna) in ER pf German hospitals and in case of inpatient admission due to other acute illnesses.

The article is interesting, well written and structured. The research methodology is relevant and I recommend this publication in International Journal of Environmental Research and Public Health with the following minor revisions.

Introduction:

-        To introduce the acronym RSV for the first utilization and to define the acronym RNA

-        Paragraph 2: However, die the utilization of RT-PCT…

-        To reinforce the choice of the comparator “conventional clinical judgement alone” ó Is the strategy implemented at the study moment? Not another test?

Methods

-        Model structure: To introduce the reference (Figure 1) after the first sentence

Results

-        The presentation of the results is clear but I would appreciate to have an additional graphic representation of univariate analyses (Tornado diagram).

-        The tables 3 and 4 should be consecutive and need to be arranged

Author Response

We are appreciative of the reviewers’ very useful input. We have revised the manuscript according to their suggestions, to the degree possible, and prepared a point-by-point response. Altered or added passages are marked red in the revised version. We hope that our manuscript now meets your expectations for publication in IJERPH.

Reviewer 3:

Introduction:

- To introduce the acronym RSV for the first utilization and to define the acronym RNA

Reply: Thank you, done!

- Paragraph 2: However, die the utilization of RT-PCT...

Reply: Thank you, corrected!

- To reinforce the choice of the comparator “conventional clinical judgement alone”  Is the strategy implemented at the study moment? Not another test?

Reply: Indeed, the Multiplex PCR test has been implemented in some German hospitals in the meantime, but to my knowledge, these hospitals do not want to be explicitly mentioned in an economic  model calculation. In order not to provoke any misunderstandings, we have therefore preferred not to mention them.

Methods

- Model structure: To introduce the reference (Figure 1) after the first sentence

Reply: Done as suggested.

Results

- The presentation of the results is clear but I would appreciate to have an additional graphic representation of univariate analyses (Tornado diagram).

Reply: Good idea, now inserted!

- The tables 3 and 4 should be consecutive and need to be arranged

Reply: Done.